# MODEL-EFFICIENT DEEP LEARNING WITH KERNELIZED CLASSIFICATION

## ABSTRACT

We investigate the possibility of using the embeddings produced by a lightweight network more effectively with a nonlinear classification layer. Although conventional deep networks use an abundance of nonlinearity for representation (embedding) learning, they almost universally use a linear classifier on the learned embeddings. This could be suboptimal for a network with a limited-capacity backbone since better nonlinear classifiers could exist in the same embedding vector space. We advocate a nonlinear kernelized classification layer for deep networks to tackle this problem. We theoretically show that our classification layer optimizes over all possible radial kernel functions on the space of embeddings to learn an optimal nonlinear classifier. We then demonstrate the usefulness of this layer in learning more model-efficient classifiers in a number of computer vision and natural language processing tasks.

## 1 INTRODUCTION

A traditional classification deep network consists of two parts: a *representation learner* that maps an input to a vector-valued representation called the embedding, and a *classifier* that classifies this embedding into the correct class. For example, in the text classification setting, the input text may be sent through a transformer encoder network with CLS-pooling (the representation learner) to obtain an embedding vector for the text (Devlin et al., 2018). A fully-connected layer (the classifier) is then operated on this embedding. A classifier learned in a usual fully connected layer with the softmax loss is linear in the space of embeddings. Therefore, the representation learner has to learn linearly-separable embeddings to do well in the classification task.

When the representation learner backbone is unable to learn linearly separable embeddings to satisfactorily solve a given classification task, the usual fix is to use a deeper and/or wider backbone to generate better embeddings (Turc et al., 2019; He et al., 2016). However, bigger backbones demand more resources in terms of compute, memory, and model storage during both training and inference. Therefore, it is natural to ask whether it is instead possible to use embeddings produced by a capacity-limited network more effectively by using a more sophisticated classification layer.

In this work we address this problem by proposing a nonlinear, kernelized classification layer. This classification layer finds an optimal *nonlinear* classifier for the embeddings by mapping them into a Reproducing Kernel Hilbert Space (RKHS) that optimally separates them into different classes. We borrow the key idea from kernel methods in the classic machine learning literature (Cortes & Vapnik, 1995; Schölkopf & Smola, 2002): instead of running a classifier directly on the embeddings, they are first mapped to much higher dimensional vectors in an RKHS using a positive definite kernel function. A linear classifier is then run on this high-dimensional RKHS. Since the dimensionality of the embeddings is dramatically increased via this mapping, a linear classifier in the RKHS corresponds to a powerful nonlinear classifier in the original embedding space. Such a classifier can therefore utilize even linearly inseparable embeddings to satisfactorily solve a classification task as demonstrated in Figure 1.

A common issue with traditional kernel methods is the choice of the kernel function used to obtain the RKHS mapping. While a collection of well-known kernels such as the linear kernel, polynomial kernels, and the Gaussian RBF kernel is available, it is often unclear which kernel would work best for a given problem. We tackle this issue by sweeping over all possible kernel functions within the deep network itself to find the optimal one via backpropagation and stochastic gradient descent.

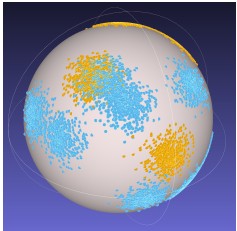 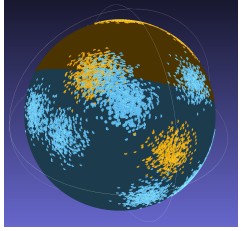 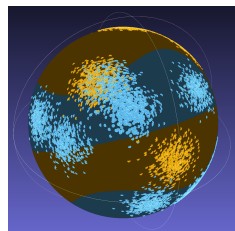

(a) Training data      (b) Softmax classifier      (b) Kernelized classifier

Figure 1: Benefits of kernelized classification. *In the second and third images, regions identified by each classifier are shown in blue and orange colors. Note that the usual softmax classifier can only separate cap-like "linear" regions on the sphere, whereas our kernelized classifier can do more complex nonlinear classification thanks to the higher dimensional RKHS embedding of the sphere. See § 6.1 for the experiment details.*

There is a wealth of literature on making deep learning models more efficient by using techniques such as pruning, quantization, low-rank factorization, and distillation (Cheng et al., 2017; Deng et al., 2020). We approach the model efficiency from an orthogonal angle by asking whether it is possible to utilize embeddings produced by a given representation learner more effectively by doing nonlinear classification. Therefore, our approach is complementary to these existing model compression techniques.

In summary, we propose a method for model-efficient classification in a deep network with three contributions: (i) we introduce a kernelized classification layer with built-in kernel learning that can utilize a given representation learning model more effectively. (ii) we theoretically establish that it is possible to optimize, within the deep network itself, over all possible kernel functions we care about to find the best RKHS that optimally separates embeddings. (iii) we empirically show that the kernelized classification layer is a viable alternative to using a larger backbone to improve the classification accuracy in a number of computer vision and natural language processing tasks.

## 2 RELATED WORK

There have been several explorations of loss functions other than the usual softmax crossentropy loss in CNN settings, specially in the open-set classification setting (Wen et al., 2019; Cevikalp & Saglamlar, 2021; Deng et al., 2019; Wang et al., 2018). An example is the Center Loss (Wen et al., 2019), which encourages low intra-class variance in the feature vectors. Other methods such as Deng et al. (2019) achieve higher performance by leaving additional margins in the softmax loss. Our work differs from these since we work in the closed-set classification setting and employ an automatically learned kernel to obtain nonlinear classification boundaries.

Second order pooling methods (Lin et al., 2015; Li et al., 2017; Wang et al., 2020) propose a way to perform nonlinear classification in the embedding space. In has been shown that second order pooling is equivalent to using a second-degree polynomial kernel (Gao et al., 2016; Cai et al., 2017). Cui et al. (2017) extended second-order pooling to higher orders while learning the coefficients of higher order interactions. Their method requires explicit calculation of feature maps, which they tackle using Fast Fourier Transforms. Mahmoudi et al. (2020), use the kernel trick in the dense layer along with the polynomial kernel. Our work differs from the above works in that we never calculate explicit feature maps and we theoretically show that our method learns over the space of all possible positive definite kernels on the hyper-sphere, which includes all polynomial kernels.

Some methods focus on extending the linear convolution in CNNs to a nonlinear operation. Convolutional kernel networks (Mairal et al., 2014) provide a kernel approximation to interpret convolutions. Zoumpourlis et al. (2017) used Volterra series approximations to extend convolutions to a nonlinear operation. Wang et al. (2019b) proposed a kernelized version of the convolution operation and demonstrated that it can learn more complicated features than the usual convolution. Our work differs from theirs in a number of ways: Some kernels used in their work, such as the $L^p$-norm kernels, are not positive definite (Berg et al., 1984) and therefore do not represent a valid RKHS mapping (Aronszajn, 1950). In contrast, we strictly work with positive definite kernels, which represent valid mappings to an RKHS. Furthermore, learning of hyperparameters of pre-defined kernels advocated in their work is principally different from the kernel learning method presented in this paper – we theoretically show that our method optimizes over the space of *all* radial positive definite kernels on the unit sphere to find the best kernel, instead of limiting the optimization to the hyperparameters of pre-defined kernels.

Prior to the dominance of deep learning methods, picking the right kernel for a given problem has been studied extensively in works such as Howley & Madden (2005); Ali & Smith-Miles (2006); Jayasumana et al. (2014); Gönen et al. (2011). In particular, Multiple Kernel Learning (MKL) approaches (Gönen et al., 2011; Varma & Ray, 2007) were popular in conjunction with SVM. Unfortunately, these methods scale poorly with the size of the training dataset. In this work, we automatically learn the kernel within a deep network. This not only allows automatic representation learning, but also scales well for large training sets. Kernels have also been considered for deep learning to reduce the memory footprint of CNNs. This was accomplished by achieving an end-to-end training of a Fastfood kernel layer (Yang et al., 2015), which uses approximations of kernel functions using Fastfood transforms (Le et al., 2013). Other related methods involving both kernels and deep learning include stochastic kernel machines (Dai et al., 2015), deep SimNets (Cohen et al., 2015), scalable deep kernels (Wilson et al., 2015), KerNET (Lauriola et al., 2020), and deep belief network based work of Le et al. (2016).

## 3 NONLINEAR SOFTMAX CLASSIFICATION

In this section we discuss the usual classification inside a deep network and its nonlinear extension. Let us consider a classification problem with a training set $\{(x_i, y_i)\}_{i=1}^N$, where each $x_i \in \mathcal{X}$, each $y_i \in [L] \doteq \{1, 2, \ldots, L\}$, $\mathcal{X}$ is a nonempty set, $L$ is the number of labels, and $N$ is the number of training examples. For instance, each training datum $(x_i, y_i)$ can be an image with its class label.

A deep neural network that solves this task has two components: a *representation learner* and a *classifier*. In the case of image classification, the representation learner consists of modules such as convolution layers, max-pooling layers, and fully-connected layers. The classifier is the last fully-connected layer operating on the learned representations (embeddings). This layer is endowed with a loss function during training.

Let $r^{(\Theta)} : \mathcal{X} \to \mathbb{R}^d$ denote the representation learner, where $d$ is the dimensionality of the embeddings and $\Theta$ represents all the parameters in this part of the network. The classifier is characterized by a function $g^{(\Omega)} : \mathbb{R}^d \to [L]$, where $\Omega$ denotes all the parameters in the last layer of the network. Usually, $\Omega$ consists of weight vectors $\mathbf{w}_1, \mathbf{w}_2, \ldots, \mathbf{w}_L$ with each $\mathbf{w}_j \in \mathbb{R}^d$, and bias terms $b_1, b_2, \ldots, b_L$ with each $b_j \in \mathbb{R}$. The function $g^{(\Omega)}$ then takes the form:

$$g^{(\Omega)}(\mathbf{z}) = \underset{j}{\operatorname{argmax}} \ \mathbf{w}_j^T \mathbf{z}, \tag{1}$$

where $\mathbf{z} = r^{(\Theta)}(x) \in \mathbb{R}^d$ is the embedding for input the $x$. Note that we have dropped the additive bias term $b_j$, with no loss of generality, to keep the notation uncluttered. During inference, the deep network's class prediction $\hat{y}^*$ for an input $x^*$ is the composite of these two functions: $\hat{y}^* = \left(g^{(\Omega)} \circ r^{(\Theta)}\right)(x^*)$.

Although conceptually there are two components of the deep network, their parameters $\Theta$ and $\Omega$ are learned jointly during training. The de facto standard way of training a classification network is minimizing the *softmax loss* applied to the classification layer. The softmax loss is the combination of the softmax function and the cross-entropy loss. More specifically, for a single training example $(x, y)$ with the embedding $\mathbf{z} = r^{(\Theta)}(x)$, the softmax loss is calculated as:

$$l(y, \mathbf{z}) = -\log\left(\frac{\exp(\mathbf{w}_y^T \mathbf{z})}{\sum_{j=1}^L \exp(\mathbf{w}_j^T \mathbf{z})}\right). \tag{2}$$

Note that the classifier $g^{(\Omega)}$ trained is this manner is completely linear in $\mathbb{R}^d$, the space of the embeddings $\mathbf{z}$s, as is evident from (1).

From the classic knowledge in kernel methods, we are aware that more powerful nonlinear classifiers on $\mathbb{R}^d$ can be obtained using the kernel trick. The key idea here is to first map the embeddings $\mathbf{z}$s into a high-dimensional RKHS $\mathcal{H}$ and perform classification there. Although the classification is linear in the high-dimensional $\mathcal{H}$, it is *nonlinear* in the original embedding space $\mathbb{R}^d$. Let $\phi : \mathbb{R}^d \to \mathcal{H}$ represent this RKHS embedding. Performing classification in $\mathcal{H}$ is then equivalent to training the neural network with the following modified version of the softmax loss:

$$l_{\mathrm{nl}}(y, \mathbf{z}) = -\log\left(\frac{\exp\left(\langle\phi(\mathbf{w}_y), \phi(\mathbf{z})\rangle_{\mathcal{H}}\right)}{\sum_{j=1}^L \exp\left(\langle\phi(\mathbf{w}_j), \phi(\mathbf{z})\rangle_{\mathcal{H}}\right)}\right), \tag{3}$$

where $\langle .,. \rangle_{\mathcal{H}}$ denotes the inner product in the Hilbert space $\mathcal{H}$. The key difference between (2) and (3) is that the dot products between $\mathbf{w}_j$s and $\mathbf{z}$ have been replaced with the inner products between $\phi(\mathbf{w}_j)$s and $\phi(\mathbf{z})$. The more general notion of *inner product* is used instead of *dot product* because the Hilbert space $\mathcal{H}$ can be infinite dimensional.

For a network trained with this nonlinear softmax function, predictions can be obtained using a modified version of the predictor:

$$g_{\mathrm{nl}}^{(\Omega)}(\mathbf{z}) = \underset{j}{\mathrm{argmax}} \; \langle \phi(\mathbf{w}_j), \phi(\mathbf{z}) \rangle_{\mathcal{H}} . \tag{4}$$

Note that the Hilbert space embeddings $\phi(.)$s can be very-high, even infinite, dimensional. Therefore, computing and storing them can be problematic. We can use the *kernel trick* from classic machine learning (Schölkopf & Smola, 2002; Shawe-Taylor & Cristianini, 2004) to overcome this problem: explicit computation of $\phi(.)$s can be avoided by directly evaluating the inner product between them using a kernel function $k : \mathbb{R}^d \times \mathbb{R}^d \to \mathbb{R}$. That is:

$$\langle \phi(\mathbf{w}), \phi(\mathbf{z}) \rangle_{\mathcal{H}} = k(\mathbf{w}, \mathbf{z}). \tag{5}$$

Intuitively, mapping $d$-dimensional embeddings into a much higher dimensional RKHS using a kernel would help in finding complex, nonlinear patterns in the embeddings that the usual softmax classification is unable to find due to its linear nature. We therefore expect kernelized classification to use embeddings provided by a given representation learner more effectively.

## 4 KERNELS ON THE UNIT SPHERE

It was shown in the previous section that, given a kernel function on the embedding space, we can obtain a nonlinear classifier in the last layer of a deep network by modifying the softmax loss function during training and the predictor during inference. However, only positive definite kernels allow this trick (Aronszajn, 1950; Berg et al., 1984). There are various choices for kernel functions in the classic machine learning literature. Some popular choices include the polynomial kernel, the Gaussian RBF kernel (squared exponential kernel), and the Laplacian kernel. Nevertheless, it is often difficult to decide the optimal kernel for a given problem. Furthermore, many of the kernels have hyperparameters that need to be manually tuned. The generally accepted solution to this problem in classic kernel methods is the MKL framework (Gönen et al., 2011), where a better kernel is learned as a linear combination of some pre-defined kernels. Unfortunately, like SVM, MKL methods do not scale well with the train set size. Furthermore, usual MKL provides no guarantee to explore all possible kernel functions to find the optimal one.

In this section, we present some theoretical results that will pave the way to define a neural network layer that can automatically learn the optimal kernel from data. By formulating kernel learning as a neural network layer, we inherit the desirable properties of deep learning, including scalability and automatic representation learning. Importantly, we show that our method can sweep over the entire space of positive definite kernels applicable to our problem setting to find the best kernel. We start the discussion with the following definition of positive definite kernels (Berg et al., 1984).

**Definition 4.1.** *Let $\mathcal{U}$ be a nonempty set. A function $k : (\mathcal{U} \times \mathcal{U}) \to \mathbb{R}$ is called a **positive definite kernel** if $k(u,v) = k(v,u)$ for all $u, v \in \mathcal{U}$ and $\sum_{j=1}^{N} \sum_{i=1}^{N} c_i c_j k(u_i, u_j) \geq 0$, for all $N \in \mathbb{N}, \{u_1, \ldots, u_N\} \subseteq \mathcal{U}$ and $\{c_1, \ldots, c_N\} \subseteq \mathbb{R}$.*

Properties of positive definite kernels have been studied extensively in mathematics (Berg et al., 1984). The following summarizes some important closure properties of this class of functions.

**Proposition 4.2.** *The family of all positive definite kernels on a given nonempty set forms a convex cone that is closed under pointwise multiplication and pointwise convergence.*

*Proof.* To intuitively understand this result, it is helpful to recall that the geometry of the family of the all positive definite kernels on a given nonempty set is closely related to that of the space of the $d \times d$ symmetric positive definite matrices, which forms a convex cone. The formal proof of this proposition can be found in Remark 3.1.11 and Theorem 3.1.12 of Berg et al. (1984). □

To simplify the problem setting, we assume that both the embeddings $\mathbf{z}$s and the weight vectors $\mathbf{w}_j$s are $L^2$-normalized. Not only this simplifies the mathematics, but also it is a practice in use

for stabilizing the training of neural networks (Yi et al., 2019; Liu et al., 2017; Hoffer et al., 2018). Due to this assumption, we are interested in positive definite kernels on the unit sphere in $\mathbb{R}^d$. From now on, we use $S^n$, where $n = d - 1$, to denote this space. We also restrict our discussion to radial kernels on $S^n$. Radial kernels, kernels that only depend on the distance between the two input points, have the desirable property of translation invariance. Furthermore, all the commonly used kernels on $S^n$, such as the linear kernel, the polynomial kernel, the Gaussian RBF kernel, and the Laplacian kernel are radial kernels. The following theorem, origins of which can be traced back to Schoenberg (1942), fully characterizes radial kernels on $S^n$.

**Theorem 4.3.** *A radial kernel $k : S^n \times S^n \to \mathbb{R}$ is positive definite for any $n$ if and only if it admits a unique series representation of the form*

$$k(\mathbf{u}, \mathbf{v}) = \alpha_{-2}[\![\langle \mathbf{u}, \mathbf{v} \rangle \in \{-1, 1\}]\!] + \alpha_{-1}([\![\langle \mathbf{u}, \mathbf{v} \rangle = 1]\!] - [\![\langle \mathbf{u}, \mathbf{v} \rangle = -1]\!]) + \sum_{m=0}^{\infty} \alpha_m \langle \mathbf{u}, \mathbf{v} \rangle^m, \quad (6)$$

*where each $\alpha_m \geq 0$, $\sum_{m=-2}^{\infty} \alpha_m < \infty$, and $[\![.]\!]$ depicts the Iversion bracket.*

*Proof.* The kernel $k_1 : S^n \times S^n \to [-1, 1] : k_1(\mathbf{u}, \mathbf{v}) = \langle \mathbf{u}, \mathbf{v} \rangle$ is positive definite on $S^n$ for any $n$ since $\sum_j \sum_i c_i c_j \langle \mathbf{u}_i, \mathbf{u}_j \rangle = \| \sum_i c_i \mathbf{u}_i \|^2 \geq 0$. Therefore, from the closure properties in Proposition 4.2, the kernel $k_m : (\mathbf{u}, \mathbf{v}) \mapsto \langle \mathbf{u}, \mathbf{v} \rangle^m$ is also positive definite for any $m \in \mathbb{N}$. Furthermore, $k_m$ is positive definite for $m = 0$ since $\sum_j \sum_i c_i c_j \langle \mathbf{u}_i, \mathbf{u}_j \rangle^0 = \| \sum_i c_i \|^2 \geq 0$.

Let us now consider the two sequences of kernels $s_{\text{odd}} = k_1, k_3, \ldots, k_{2m+1}, \ldots$ and $s_{\text{even}} = k_2, k_4, \ldots, k_{2m}, \ldots$. Since $-1 \leq \langle \mathbf{u}, \mathbf{v} \rangle \leq 1$, it is clear that $s_{\text{odd}}$ and $s_{\text{even}}$ converge pointwise to $k_{\text{odd}}(\mathbf{u}, \mathbf{v}) = [\![\langle \mathbf{u}, \mathbf{v} \rangle = 1]\!] - [\![\langle \mathbf{u}, \mathbf{v} \rangle = -1]\!]$ and $k_{\text{even}}(\mathbf{u}, \mathbf{v}) = [\![\langle \mathbf{u}, \mathbf{v} \rangle \in \{-1, 1\}]\!]$, respectively. From the last closure property of Proposition 4.2, both $k_{\text{odd}}$ and $k_{\text{even}}$ are positive definite on $S^n$. Invoking Proposition 4.2 again, we conclude that any finite conic combination of the kernels $k_{\text{even}}, k_{\text{odd}}, k_0, k_1, \ldots$ is positive definite on $S^n$ for any $n$. This completes the forward direction of the proof. The proof of the converse is found in Chapter 5 of Berg et al. (1984). $\square$

Equipped with a complete characterization of the positive definite radial kernels on $S^n$, we now discuss how we can combine this result with the nonlinear softmax formulation in § 3 to automatically learn the best kernel classifier within a deep network.

## 5 THE KERNELIZED CLASSIFICATION LAYER

We now introduce a kernelized classification layer that acts as a drop-in replacement for the usual softmax classification layer in a deep network. This new layer classifies embeddings in a high-dimensional RKHS while automatically choosing the optimal positive definite kernel that enables the mapping into the RKHS. As a result, we do not have to hand-pick a kernel or its hyperparameters. Our nonlinear classification layer can utilize embeddings more efficiently than a softmax classifier, which is linear in the embedding space. We also show that the kernelized classification layers comes with negligible added cost during both training and inference.

### 5.1 MECHANICS OF THE LAYER

Our classification layer is parameterized by the usual weight vectors: $\mathbf{w}_1, \mathbf{w}_2, \ldots, \mathbf{w}_L$, and some additional learnable coefficients: $\alpha_{-2}, \alpha_{-1}, \ldots, \alpha_M$, where $M \in \mathbb{N}$ and each $\alpha_m \geq 0$. During training, this classifier maps embeddings $\mathbf{z}$s into a high-dimensional RKHS $\mathcal{H}_{\text{opt}}$ that optimally separates embeddings belonging to different classes, and learns a linear classifier in $\mathcal{H}_{\text{opt}}$. During inference, the classifier maps embeddings of previously unseen inputs to the RKHS it learned during training and performs classification in that space. This is achieved by using the nonlinear softmax loss defined in (3) during training and the nonlinear predictor defined in (4) during testing, with the inner product in $\mathcal{H}$ given by: $\langle \phi(\mathbf{w}), \phi(\mathbf{z}) \rangle_{\mathcal{H}} = \langle \phi(\mathbf{w}), \phi(\mathbf{z}) \rangle_{\mathcal{H}_{\text{opt}}} = k_{\text{opt}}(\mathbf{w}, \mathbf{z})$, where $k_{\text{opt}}(., .)$ is the reproducing kernel of $\mathcal{H}_{\text{opt}}$. The optimal RKHS $\mathcal{H}_{\text{opt}}$ for a given classification problem is learned by finding the optimal kernel $k_{\text{opt}}$ during training as discussed in the following.

Theorem 4.3 states that any positive definite radial kernel on $S^n$ admits the series representation shown in (6). Therefore, the optimal kernel $k_{\text{opt}}$ must also have such a series representation. We approximate this series with a finite summation by cutting off the terms beyond the order $M$:

$$k_{\text{opt}}(\mathbf{w}, \mathbf{z}) \approx \alpha_{-2} k_{\text{even}}(\mathbf{w}, \mathbf{z}) + \alpha_{-1} k_{\text{odd}}(\mathbf{w}, \mathbf{z}) + \sum_{m=0}^{M} \alpha_m k_m(\mathbf{w}, \mathbf{z}), \quad (7)$$

where, $k_{\text{even}}, k_{\text{odd}}, k_0, k_1, \ldots, k_M$ have meanings defined § 4 and $\alpha_{-2}, \alpha_{-1}, \ldots, \alpha_M \geq 0$. Using Proposition 4.2 and the discussion in the proof of Theorem 4.3, one can easily verify that this approximation does not violate the positive definiteness of $k_{\text{opt}}$. A rigorous analysis of the accuracy of this approximation is provided in Appendix E.

With this, $k_{\text{opt}}$ is learned automatically from data by making the coefficients $\alpha_{-2}, \alpha_{-1}, \ldots, \alpha_M$s learnable parameters of the classification layer. Let $\boldsymbol{\alpha} = [\alpha_{-2}, \alpha_{-1}, \ldots, \alpha_M]^T$. The gradient of the loss function with respect to $\boldsymbol{\alpha}$ can be calculated via the backpropagation algorithm using (3) and (7). Therefore, it can be optimized along with $\mathbf{w}_1, \mathbf{w}_2, \ldots, \mathbf{w}_L$ during the gradient descent based optimization of the network. This procedure is equivalent to automatically finding the RKHS that optimally separates the embeddings belonging to different classes.

The constraint $\alpha_{-2}, \alpha_{-1}, \ldots, \alpha_M \geq 0$ in (7) can be imposed with a ReLU function (see Appendix D.5 for more discussion). As shown later in § 7.3, the exact value of $M$ is not critical as long as it is sufficiently large. This is because, as discussed in the proof of Theorem 4.3, the higher order terms that are truncated approach either $k_{\text{odd}}$ or $k_{\text{even}}$, both of which are already included in the finite summation. On the other hand, if the terms beyond some order $M' < M$ are not important, the network can automatically learn to make the corresponding $\alpha$ coefficients vanish. We observed that $M = 10$ works well enough in practice and stick to this number in all our experiments.

Importantly, the kernelized classification layer described above can pass on the loss gradients to its inputs. Therefore, the kernelized classification layer is fully compatible with end-to-end training and can act as a drop-in replacement for an existing softmax classification layer.

## 5.2 Additional Complexity

Since we propose kernelized classification as a replacement for the usual softmax classification to improve model efficiency, one might wonder about the added cost of the kernelized classification layer. It uses $(M + 3)$ extra learnable parameters. During both training and inference, the added computational complexity is $\mathcal{O}(M + d)$ per datum, assuming a commonly-available constant-time operation for taking powers. Additional memory footprint is $\mathcal{O}(M)$ during training to account for cached gradients. Note that $M = 10$ and $d$ ranges from 64 to 768 in our experiments. Therefore, the kernelized classification comes with a negligible added cost over the softmax classification in terms of compute, memory, and trained model storage.

## 6 Experiments

We now present experimental evaluation of our method. For all experiments, the main baseline is the standard softmax classifier. Where appropriate, we show three additional baseline results based on the linear kernel, second order pooling (Lin et al., 2015), and kervolutional networks (Wang et al., 2019b). Note that the focus of our experiments is to demonstrate the benefits of kernelized classification in efficiently utilizing embeddings learned with various representation learners, not to claim state-of-the-art results on already well-explored benchmark datasets. Details about our experimental setup is in Appendix A.

### 6.1 Synthetic Data

We first evaluate the kernelized classification layer as an isolated unit by demonstrating its capabilities to learn nonlinear patterns on $S^n$. We train a softmax classifier and our kernelized classifier on the synthetic dataset described in Appendix B. Results on the test set are shown in Table 1. We also report the theoretical maximum accuracy, the accuracy of the Bayes optimal classifier. The accuracy of our kernelized classification layer significantly outperforms the baseline and gets close to theoretical best. This can be attributed to the layer's capabilities to learn nonlinear patterns on the sphere by embedding the data into an RKHS that optimally separates the classes.

We visualize the outcomes of the classifiers in Figure 1. Note that the softmax classifier can only separate cap-like regions on $S^2$, this is a result of its being a linear classifier with respect to the embeddings. Our kernelized classifier, on the other hand, can do a more complex nonlinear separation of the embeddings.

| Method | Accuracy |
|---|---|
| Softmax classifier (baseline) | 85.51 |
| Kernelized classifier (ours) | **94.20** |
| Bayes optimal classifier | 95.06 |

Table 1: Classification results on the synthetic dataset.

| Dataset | Accuracy | |
|---|---|---|
| | Baseline | Ours |
| CIFAR-10 | 76.06 | **79.85** |
| CIFAR-100 | 44.38 | **46.48** |

Table 2: Results in the distillation setting.

## 6.2 IMAGE CLASSIFICATION

We now report results on CIFAR-10 and CIFAR-100 real world image benchmarks[1] (Krizhevsky, 2009). To demonstrate better model-efficiency with the kernelized classification, we experiment with several CIFAR-ResNet architectures (He et al., 2016) with increasing model capacity. We consider four different baselines: (1) Softmax: the standard softmax loss, (2) LIN: normalized embeddings and weights with only the linear kernel along with a learnable coefficient. This is similar to the approach discussed in Hoffer et al. (2018), but with additional freedom to learn the weight vectors, (3) SOP: second order pooling (Lin et al., 2015), which is also equivalent to Mahmoudi et al. (2020) with a second degree polynomial, and (4) KERVO: kervolutional networks (Wang et al., 2019b) with the best out of the Gaussian RBF kernel and the polynomial kernel.

Increased model-efficiency obtained with the kernelized classifier is evident from the accuracies summarized in Table 3. For example, the same accuracy of ResNet-56 (540K parameters) with the standard softmax classifier can be obtained with a much smaller ResNet-32 (300K parameters) when the kernelized classifier is used. Our method significantly outperforms the other baselines as well. This shows the benefits of optimizing over the entire space of positive definite kernels instead of restricting ourselves to linear methods or pre-defined kernels.

| Backbone | # params | Softmax | LIN | SOP | KERVO | Ours |
|---|---|---|---|---|---|---|
| ResNet-8 | 61K | 83.73 / 53.82 | 82.45 / 54.00 | 84.03 / 55.80 | 85.15 / 56.92 | **86.93 / 58.27** |
| ResNet-14 | 121K | 89.87 / 63.85 | 90.16 / 63.67 | 90.47 / 63.53 | 90.43 / 64.14 | **91.48 / 66.67** |
| ResNet-20 | 181K | 91.14 / 65.99 | 91.01 / 65.79 | 91.75 / 67.97 | 91.34 / 67.31 | **92.88 / 69.33** |
| ResNet-32 | 300K | 92.22 / 68.96 | 92.21 / 69.16 | 92.31 / 70.39 | 92.42 / 69.40 | **93.70 / 71.30** |
| ResNet-44 | 420K | 92.10 / 70.16 | 93.10 / 70.54 | 92.42 / 71.13 | 92.88 / 71.09 | **94.05 / 73.20** |
| ResNet-56 | 540K | 93.01 / 71.23 | 93.13 / 72.11 | 93.33 / 73.12 | 93.10 / 72.39 | **94.15 / 74.10** |

Table 3: Results on the CIFAR-10/CIFAR-100 datasets.

## 6.3 NATURAL LANGUAGE UNDERSTANDING

In this section, we show the benefits of the kernelized classification layer in solving four different text classification tasks in the GLUE benchmark (Wang et al., 2019a). We use mask-LM pretrained BERT models of different capacities (Turc et al., 2019) and finetune them on each classification task. Note that we do not use distillation and directly finetune the models with the dataset labels. Since detailed analyses on GLUE test datasets are not allowed (Wang et al., 2019a), we tune hyperparameters on subsets of train sets and report results on the validation sets in Table 4. More details about the experiment setup are provided in Appendix A.

Table 4 provides evidence that kernelized classification layers helps in extracting more gains out of a given representation learner. For example, across all the datasets, BERT-Mini (11.3M parameters) with kernelized classification can get similar results as the BERT-Small (29.1M parameters) with softmax classification. Therefore, using the nonlinear, kernelized classifier is an effective alternative to using a bigger backbone for increasing classification performance.

## 6.4 KNOWLEDGE DISTILLATION

We now evaluate our method in the distillation setting to show that it is complementary to existing model compression techniques. We used the CIFAR-10 and CIFAR-100 datasets, the softmax CIFAR ResNet-56 models from Tables 3 as the teacher models, and the LeNet-5 network (Lecun et al., 1998) as the student model. Note that it is straightforward to utilize the kernelized classification layer in the distillation setting described in Hinton et al. (2015) by replacing usual logits with their

---

[1]We use the standard data augmentation in CIFAR-10/100 (He et al., 2016). Some authors refer to these datasets as CIFAR-10+/100+ when data augmentation is used. We omit the + to keep the text uncluttered.

| Dataset | Method | BERT-Tiny | BERT-Mini | BERT-Small | BERT-Medium | BERT-Base |
|---|---|---|---|---|---|---|
| # params | | 4.4M | 11.3M | 29.1M | 41.7M | 110.1M |
| MPRC ($F_1$/Acc) | Softmax | 82.77 / 74.50 | 85.20 / 79.25 | 88.41 / 83.75 | 89.72 / 85.50 | 92.58 / 89.50 |
| | LIN | 82.00 / 73.78 | 84.99 / 79.10 | 88.51 / 83.79 | 89.88 / 85.62 | 92.34 / 89.48 |
| | SOP | 83.81 / 76.21 | 86.21 / 80.42 | 89.21 / 84.38 | 89.99 / 85.78 | 92.68 / 89.73 |
| | Ours | **84.97 / 78.25** | **88.89 / 84.50** | **90.12 / 85.75** | **91.60 / 87.75** | **93.24 / 90.50** |
| QQP ($F_1$/Acc) | Softmax | 81.91 / 86.35 | 84.09 / 88.10 | 85.18 / 88.87 | 86.12 / 89.73 | 87.18 / 90.30 |
| | LIN. | 81.07 / 86.11 | 84.05 / 88.09 | 85.19 / 88.87 | 86.05 / 89.69 | 87.20 / 90.31 |
| | SOP | 82.01 / 87.00 | 84.50 / 88.56 | **85.99 / 90.00** | 86.15 / 89.99 | 87.21 / 90.34 |
| | Ours | **82.78 / 87.13** | **85.00 / 88.84** | 85.83 / 89.46 | **86.83 / 90.17** | **87.97 / 91.04** |
| RTE (Acc) | Softmax | 63.53 | 65.74 | 66.01 | 66.84 | 71.46 |
| | LIN | 62.32 | 65.69 | 65.97 | 66.02 | 71.50 |
| | SOP | 63.59 | 65.78 | 66.00 | **68.02** | 72.96 |
| | Ours. | **65.96** | **66.93** | **67.72** | 67.94 | **73.28** |
| SST-2 (Acc) | Softmax | 82.38 | 87.11 | 87.19 | 89.86 | 91.97 |
| | LIN. | 82.04 | 86.92 | 87.29 | 89.50 | 91.72 |
| | SOP | 82.89 | **87.19** | 88.52 | 90.12 | 91.98 |
| | Ours | **84.07** | 87.14 | **89.42** | **90.83** | **92.69** |

Table 4: Results on several natural language understanding tasks in the GLUE benchmark.

kernelized counterparts. We use the cross-entropy loss with the teacher scores with the temperature set to 20 in all cases. Results are shown in Table 2. Significant gains are observed with the kernelized classification layer. This can be attributed to the layer's capabilities to approximate complex teacher probabilities even with weak embeddings due to the nonlinear classifier.

## 6.5 ACTIVE LEARNING

Active learning focuses on reducing human annotation costs by selecting a subset of images to label that are more likely to yield the best model (Settles, 2009). We used different sampling methods such as random, margin (Lewis & Gale, 1994; Scheffer et al., 2001), and k-center (Sener & Savarese, 2017; Wolf, 2011) to generate subsets of various sizes. The setup is detailed in Appendix C. As shown in Table 5, our results on random subsets outperform the softmax ResNet-56 models on margin and k-center based subsets, and we achieve even better results using improved sampling methods. This demonstrates that the kernelized classification layer can produce better models in limited-data settings as well.

## 7 ABLATION STUDIES

We now present ablation experiments using CIFAR-100 and the ResNet-56 backbone.

### 7.1 KERNEL LEARNING

To investigate the benefits of automatic kernel learning compared to using a pre-defined kernel in the kernelized classification layer, we compare our kernel learning method with two pre-defined kernels in the kernelized classification layer: the polynomial kernel of order 10 and the Gaussian RBF kernel. We also show results with an MKL baseline with linear, 2nd order, and Gaussian

| % | Baseline | | | Ours | | |
|---|---|---|---|---|---|---|
| | rnd | mgn | k-ctr | rnd | mgn | k-ctr |
| 30 | 58.03 | 58.88 | 58.41 | 61.66 | 61.80 | **63.08** |
| 40 | 61.05 | 61.81 | 62.02 | 65.25 | 66.28 | **66.35** |
| 50 | 64.81 | 65.36 | 65.47 | 67.17 | 68.14 | **69.41** |
| 60 | 66.26 | 67.03 | 68.25 | 69.17 | **70.61** | 70.10 |
| 70 | 67.47 | 69.16 | 69.84 | 70.06 | 70.90 | **71.50** |
| 80 | 69.59 | 69.47 | 71.25 | 71.66 | 72.21 | **72.64** |
| 90 | 70.25 | 71.41 | 71.11 | 72.60 | **73.90** | 73.14 |

Table 5: Active learning on CIFAR-100. Terms rnd, mgn, and k-ctr refer to random, margin, and k-center, respectively.

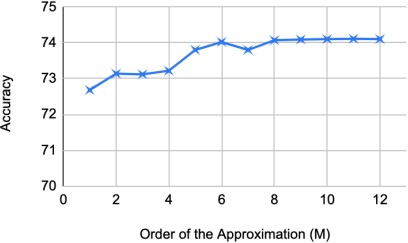

Figure 2: Accuracy versus the order of the approximation ($M$) on CIFAR-100.

kernels. Results are shown in Table 6. It is evident that sweeping over all possible kernels in light of Theorem 4.3 to find the best kernel yields significant practical benefits. This is not surprising because the polynomial and Gaussian kernels are members of the large search space of kernels that we optimize over in our method.

| Method | Acc |
|---|---|
| Gaussian RBF with fixed $\sigma$ | 73.21 |
| Gaussian RBF with learned $\sigma$ | 73.23 |
| Polynomial kernel | 73.16 |
| MKL | 73.25 |
| Ours | **74.10** |

Table 6: Benefits of learning the best kernel.

| Method | Acc |
|---|---|
| No normalization | 71.16 |
| Embeddings normalized | 71.23 |
| Embeddings & weights normalized with fixed $T$ | 71.19 |
| Embeddings & weights normalized with learned $T$ | 72.11 |
| Ours | **74.10** |

Table 7: Effects of normalization.

## 7.2 EMBEDDING NORMALIZATION

As discussed previously, we $L^2$-normalize both embeddings and weights in the classification layer. We study the effect of this normalization for the baseline softmax loss in Table 7. Note the setting where both embeddings and weights are normalized with an appropriate temperature $T$ is equivalent to the cosine softmax loss (Liu et al., 2017; Chen et al., 2019; Wang et al., 2020). Note also that the LIN baseline considered in Tables 3 and 4 uses normalized embeddings and weights with an automatically learned $T$, same as the fourth row in Table 7. In our softmax baselines we use normalized embeddings since it works consistently better than the unnormalized version.

## 7.3 SENSITIVITY TO THE ORDER OF THE APPROXIMATION

As discussed in § 5.1, intuitively, the order $M$ of the approximation should not matter as long as it is large enough. We verify this in Figure 2 with the CIFAR-100 dataset, where we show the changes in accuracy with increasing $M$. We use $M = 10$ in all our experiments.

## 7.4 DO MORE FULLY-CONNECTED LAYERS HELP?

One could wonder whether more fully-connected layers at the end of the network would improve the classification. To address this, we added an additional fully-connected layer with $d$ units to ResNet-56. The accuracy improved only marginally from 71.23 to 71.29, as opposed to 74.10 with our method. This observation is in-line with the discussion in Rendle et al. (2020): MLP scorers are somewhat difficult to train. An explanation for this could be the common observation that, although MLPs can theoretically approximate any function, learning one from data is difficult. This has motivated inductive-bias based models such as CNNs and Transformers. Kernelized classification can also be viewed as a way of presenting an inductive bias motivated by the classic kernel method theory. Note also that, unlike the kernelized classification layer, added MLP layers come with a significant increase in the model's computational and memory complexity.

## 7.5 DIFFERENT LOSS FUNCTIONS, ACTIVATION FOR COEFFICIENT, ETC.

We also provide a number of other ablations studies on different loss functions, effects of embedding rectification, kernel coefficient activation, and other settings in Appendix D.

## 8 CONCLUSION

We presented a kernelized classification layer for deep neural networks aiming to extract the best possible classifier with embeddings produced by a given representation learner. This classification layer classifies embeddings in a high dimensional RKHS while automatically learning the optimal kernel that enables this high-dimensional mapping. We showed that a classification network with a lightweight representation learning backbone can be made more effective by replacing the usual softmax classifier with the kernelized classifier. We showed consistent and substantial accuracy improvements in image classification, natural language understanding, distillation, and active learning settings. These accuracy improvements strongly support the usefulness of kernelized classification layer in finding nonlinear patterns in the embeddings.

## ETHICS STATEMENT

This work concerns mathematical and empirical analysis of deep learning based classification techniques with applications in image recognition and natural language understanding. We do not foresee our work having undue societal effects. Our work does not explicitly consider issues of fairness in classification, which is an important yet under-studied dimension. We do not foresee our techniques as unduly amplifying biases in existing algorithms.

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

## A    EXPERIMENTAL SETUP

In all experiments, we use $M = 10$ for the kernelized classification layer. As discussed in § 5.1 and Appendix D.5, we use $\mathrm{ReLU}$ activation and a weight decay of $0.0001$ on the $\boldsymbol{\alpha}$ parameter vector. This is the same amount of weight decay used in the other parts of the network, when applicable. The $\boldsymbol{\alpha}$ parameter vector is initialized with all ones.

For all image classification experiments, we use SGD with 0.9 momentum, linear learning rate warmup (Goyal et al., 2017), cosine learning rate decay (Loshchilov & Hutter, 2017), and decide the base learning by cross validation. When a better learning rate schedule is available for the baseline (e.g. the CIFAR schedule in He et al. 2016), we experimented with both that and our schedule and report the best accuracy of the two. The maximum number of epochs was 450 in all cases. Mini-batch size was 128 for the synthetic and CIFAR datasets and 64 other datasets with larger images used in Appendix D.2. We used the standard CIFAR data augmentation method in He et al. 2016 for CIFAR-10 and CIFAR-100 datasets, and the Imagenet data augmentation in the same paper for other image datasets. Some ResNet models use $\mathrm{ReLU}$ activation on embeddings. We also remove this activation to utilize the full surface of $S^n$. The same is done to the baseline models to enable a fair comparison (see AppendixD.4 for more details).

For the natural language understanding tasks in § 6.3, we use the publicly-available, MLM-pretrained BERT/Small-BERT models from TensorFlow Hub (Turc et al., 2019). Usual CLS pooling is done at the end of the Transformer encoder to obtain an embedding for each input sentence/sentence pair. We use the AdamW optimizer with linear ramp-up and decay as is standard for BERT model finetuning. Note that we do not distill the final task from a bigger model and directly finetune with the one-hot labels in the original datasets. Hyperparameter search followed the procedure described in Turc et al. (2019). Since detailed analysis of different methods on GLUE test tests are not allowed, we tune hyperparameters on a 10% subset of the train set and report results on the validation sets. Once the hyperparamers are decided, the full train set is used to train the final model. Since GLUE validation sets are small, we report the average accuracy over 5 different runs for each method. We do not however notice significant variance in accuracy across different runs.

## B    SYNTHETIC DATA GENERATION

We generated the binary classification dataset used in §6.1 using a mixture of Gaussians, inspired by the blue-orange dataset in Hastie et al. (2001). More specifically, we first generated 10 cluster centers for each class by sampling from an isotropic Gaussian distribution with covariance $0.5\,I_3$ and mean $[1, 0, 0]^T$ for the blue class and $[0, 1, 0]^T$ for the orange class. We then generated 5,000 train observations for each class using the following method: for each observation, we uniform-randomly picked a cluster center of the corresponding class and then generated a sample from an isotropic Gaussian distribution centered at that cluster center with covariance $0.02\,I_3$. All the observations were projected on to $S^2$ by $L^2$-normalizing them. The test set was generated in the same manner using the same cluster centers as the train set.

## C    ACTIVE LEARNING

The goal of the active learning experiment is to show that the kernelized classification layer can produce accurate models even with less data. In order to study this, we produce subsets of datasets under various budgets using several sampling techniques, and evaluate the models trained on this. The simplest one is random sampling, where images are selected randomly under a given budget. Other methods include margin (Lewis & Gale, 1994; Scheffer et al., 2001), and k-center (Sener & Savarese, 2017; Wolf, 2011) where class prediction scores and embeddings from the images are used in the subset selection.

We do not rely on the actual labels of the images in the dataset during the subset selection, since active learning is driven toward reducing label annotation costs. We used a 10% random subset of the original CIFAR-100 dataset with labels to first learn an initial seed model, which was then used to generate predictions and embeddings. Note that only embeddings and class prediction scores from this initial seed model are used in subset selection, and we do not access the original class labels of the images. We use a batch setting where we do not incrementally update the model after selecting

every image, and we directly select entire subsets under a given budget. In all our experiments, we used the CIFAR ResNet-56 model. The learning rate, batch size and the number of epochs are provided in Appendix A. The embeddings are of dimension 64. For the k-center method, we need distances between the embeddings, and we used cosine distances computed using fast similarity search of (Guo et al., 2020).

# D   ADDITIONAL EXPERIMENTS

We report a number of additional experimental results in this section.

## D.1   ADDITIONAL BACKBONES

Image classification results with VGG-16 and DenseNet-40-12 backbones are reported in Table 8. Since the original VGG-16 is designed for $224 \times 224$ images, we used a modified CIFAR version with 256 dimensional hidden size at the end.

| Backbone | Softmax | LIN | SOP | KER | Ours |
|---|---|---|---|---|---|
| VGG-16 | 92.58 / 71.48 | 92.59 / 71.52 | 93.20 / 72.10 | 93.21 / 72.04 | **94.39 / 73.32** |
| DenseNet | 94.76 / 75.58 | 94.73 / 75.67 | 94.98 / 75.00 | 94.92 / 75.08 | **95.31 / 76.87** |

Table 8: Results on the CIFAR-10/CIFAR-100 datasets with different backbones.

## D.2   IMAGE CLASSIFICATION TRANSFER LEARNING

Here, we evaluate our method in a image classification transfer learning setting. To this end, we take a ResNet-50 network pre-trained on the Imagenet ILSVRC 2012 classification dataset (Deng et al., 2009) and finetune it on Oxford-IIIT Pets (Parkhi et al., 2012) and Stanford Cars (Krause et al., 2013) datasets. For each dataset, we use the train/test splits provided by the standard Tensorflow Datasets implementation (Authors). Results are summarized in Table 9. Note that the KERVO baseline is not possible in this setting as it involves changes to the backbone network. On both datasets, kernelized classification layer results in significant gains over the baselines. This is intuitive to understand since the embeddings learned from the source task (Imagenet) might not linearly separate the new classes in the target task. We can therefore benefit from a nonlinear classifier in the transfer learning setting.

| Dataset | Accuracy | | | |
|---|---|---|---|---|
| | SM | LIN | SOP | Ours |
| Oxford-IIIT Pets | 92.06 | 91.99 | 92.28 | **93.56** |
| Stanford Cars | 90.78 | 90.83 | 91.04 | **92.60** |

Table 9: Results on the transfer learning datasets.

## D.3   DIFFERENT LOSS FUNCTIONS

To evaluate the kernelized classification layer under a loss function other than the cross-entory loss, we report CIFAR-10/100 results with the squared loss (Hui & Belkin, 2020) in Table 10. Note that the application of squared loss to the kernelized classification layer's outputs is straightforward since it outputs logits in the usual sense.

## D.4   EFFECT OF EMBEDDING RECTIFICATION

As discussed previously, different to the usual image classification networks in He et al. (2016), we remove the ReLU activation from the embeddings. This is to utilize the full surface of $S^n$ without restricting ourselves to only the nonnegative orthant. As is evident from Table 11, removing ReLU has only a marginal effect on the standard softmax baseline. It is however an important factor for our method. We consistently used embeddings without the ReLU activation in all our experiments in the previous sections.

| Backbone | CIFAR-10 | | CIFAR-100 | |
|---|---|---|---|---|
| | Sq. Loss | Ours+Sq.Loss | Sq. Loss | Ours+Sq. Loss |
| ResNet-8 | 83.70 | **86.58** | 51.55 | **55.99** |
| ResNet-14 | 89.86 | **91.67** | 62.94 | **65.06** |
| ResNet-20 | 91.20 | **92.75** | 64.00 | **68.61** |
| ResNet-32 | 92.19 | **93.65** | 68.10 | **71.35** |
| ResNet-44 | 92.16 | **94.12** | 69.48 | **72.25** |
| ResNet-56 | 93.19 | **94.22** | 70.34 | **73.13** |

Table 10: Results on the CIFAR-10/100 datasets with the square loss.

| Method | Accuracy |
|---|---|
| Softmax classifier with: | |
|     rectified embeddings | 70.96 |
|     unrectified embeddings | 71.23 |
| Our classifier with: | |
|     rectified embeddings | 71.61 |
|     unrectified embeddings | **74.10** |

Table 11: Effect of rectification of the embeddings.

### D.5 EFFECT OF ACTIVATION ON THE KERNEL COEFFICIENTS

Following the discussion in § 5.1, the constraint $\alpha_{-2}, \alpha_{-1}, \ldots, \alpha_M \geq 0$ is important to preserve the positive definiteness of $k_{\mathrm{opt}}$. This can be imposed by using $\boldsymbol{\alpha} = \mathrm{ReLU}(\boldsymbol{\alpha}')$, where $\boldsymbol{\alpha}'$ is the learnable parameter vector. However, ReLU has no upper-bound and allowing the scale of $\boldsymbol{\alpha}$ to grow unboundedly causes issues in optimization: Assume we have an instantiation $\boldsymbol{\alpha}_0$ of the vector $\boldsymbol{\alpha}$. By replacing $\boldsymbol{\alpha}_0$ with $\lambda\boldsymbol{\alpha}_0$, where $\lambda > 1$, we scale all the inner product terms in (3) and (4) by the same $\lambda$. As a result, we improve the loss of the already correctly classified training examples, but without making any effective change to the predictor. Therefore, under this setting, once the majority of the training examples are correctly classified, the neural network can easily improve the loss just by increasing the norm of $\boldsymbol{\alpha}$, which is not useful. We therefore advocate an $L^2$-regularization term on $\boldsymbol{\alpha}$ when ReLU activation is used.

Alternatively, one could also use $\boldsymbol{\alpha} = \mathrm{sigmoid}(\boldsymbol{\alpha}')$ or $\boldsymbol{\alpha} = \mathrm{softmax}(\boldsymbol{\alpha}')$, both of which not only guarantee $\alpha_{-2}, \alpha_{-1}, \ldots, \alpha_M \geq 0$, but also produce bounded $\boldsymbol{\alpha}$. Therefore, no regularization on $\boldsymbol{\alpha}$ is needed for these options. The $\mathrm{softmax}$ activation here should not be confused with the softmax loss discussed in § 3. The usage of the $\mathrm{softmax}$ activation in this context is similar to that in the self-attention literature (Vaswani et al., 2017), where it is used to normalize the coefficients of a linear combination. We experimented with these different activations on $\boldsymbol{\alpha}'$ and summerized the results in Table 12.

We used a weight decay of 0.0001 on the coefficient vector whenever ReLU activation is used. Although $\mathrm{sigmoid}$ and $\mathrm{softmax}$ activations eliminate the need for weight decay, they put a hard constraint on $|\langle \phi(\mathbf{w}), \phi(\mathbf{f}) \rangle_{\mathcal{H}}|$. To overcome this limitation, it is helpful to use a temperature hyperparameter in (3), where each inner product is divided by $T$ before taking the exponential. We used a temperature of 0.1 and 0.005, with $\mathrm{sigmoid}$ and $\mathrm{softmax}$, respectively. Although $\mathrm{sigmoid}$ gives the best performance in Table 12, we occasionally observed optimization issues with it, which could be due to the vanishing gradient issue associated with this activation function. We therefore stick to ReLU in all other experiments. We however note that, in most cases, competitive results can be obtained with $\mathrm{softmax}$ as well, when used with a temperature of 0.005.

It is also interesting to note that using no activation function on $\boldsymbol{\alpha}'$ causes frequent divergence in training. This is consistent with the theory: The summation in (7) is not guaranteed to be positive definite when $a_m$s are allowed to be negative (see Proposition 4.2). Therefore, the theory of kernelized classification is not valid in this case.

| Activation function | Accuracy |
|---|---|
| sigmoid | **74.96** |
| softmax | 73.69 |
| ReLU | 74.10 |
| None (linear) | unstable |

Table 12: Different activation functions on the coefficient vector. Note that the kernelized classifier is unstable when no activation function is used, this agrees with the theoretical analysis.

# E    APPROXIMATION ERROR BOUNDS

In this section, we analyze error bounds for the approximation in Eq. (7). We start by proving the following theorem, which establishes a rigorous upper bound for the approximation error.

**Theorem E.1.** *Let $k : S^n \times S^n \to \mathbb{R}$ be any positive definite radial kernel on $S^n$ with the series expansion $k(\mathbf{u}, \mathbf{v}) = \alpha_{-2}k_{\text{even}}(\mathbf{u}, \mathbf{v}) + \alpha_{-1}k_{\text{odd}}(\mathbf{u}, \mathbf{v}) + \sum_{m=0}^{\infty} \alpha_m \langle \mathbf{u}, \mathbf{v} \rangle^m$, and $k_M : S^n \times S^n \to \mathbb{R}$ be its $M^{th}$ partial sum. Define $\psi : [-1, 1] \to \mathbb{R}$ as $\psi(x) := \sum_{m=0}^{\infty} \alpha_m x^m$. Then the approximation error bound for the partial sum of the kernel is given by*

$$|k(\mathbf{u}, \mathbf{v}) - k_M(\mathbf{u}, \mathbf{v})| \leq \frac{1}{(M+1)!} \max_{x \in (-1,1)} |\psi^{(M+1)}(x)|,$$

*for all $(\mathbf{u}, \mathbf{v}) \in S^n \times S^n$.*

*Proof.* Since $k$ is positive definite, from Theorem 4.3, it has a series expansion of the form:

$$k(\mathbf{u}, \mathbf{v}) = \alpha_{-2}[\![\langle \mathbf{u}, \mathbf{v} \rangle \in \{-1, 1\}]\!] + \alpha_{-1}([\![\langle \mathbf{u}, \mathbf{v} \rangle = 1]\!] - [\![\langle \mathbf{u}, \mathbf{v} \rangle = -1]\!]) + \sum_{m=0}^{\infty} \alpha_m \langle \mathbf{u}, \mathbf{v} \rangle^m$$

$$= \alpha_{-2}[\![\langle \mathbf{u}, \mathbf{v} \rangle \in \{-1, 1\}]\!] + \alpha_{-1}([\![\langle \mathbf{u}, \mathbf{v} \rangle = 1]\!] - [\![\langle \mathbf{u}, \mathbf{v} \rangle = -1]\!]) + \psi(\langle \mathbf{u}, \mathbf{v} \rangle).$$

Note that $\psi$ is an analytic function. Furthermore,

$$k_M(\mathbf{u}, \mathbf{v}) = \alpha_{-2}[\![\langle \mathbf{u}, \mathbf{v} \rangle \in \{-1, 1\}]\!] + \alpha_{-1}([\![\langle \mathbf{u}, \mathbf{v} \rangle = 1]\!] - [\![\langle \mathbf{u}, \mathbf{v} \rangle = -1]\!]) + \sum_{m=0}^{M} \alpha_m \langle \mathbf{u}, \mathbf{v} \rangle^m$$

$$= \alpha_{-2}[\![\langle \mathbf{u}, \mathbf{v} \rangle \in \{-1, 1\}]\!] + \alpha_{-1}([\![\langle \mathbf{u}, \mathbf{v} \rangle = 1]\!] - [\![\langle \mathbf{u}, \mathbf{v} \rangle = -1]\!]) + \psi_M(\langle \mathbf{u}, \mathbf{v} \rangle),$$

where $\psi_M(x) := \sum_{m=0}^{M} \alpha_m x^m$. Therefore, $\psi_M(x)$ is the $M^{\text{th}}$ order Maclaurin polynomial approximation of $\psi(x)$. Since $\psi(x)$ is analytic, and therefore infinitely differentiable, we can obtain the Lagrange form of the approximation error as:

$$\psi(x) - \psi_M(x) = \frac{\psi^{(M+1)}(\xi)}{(M+1)!} x^{(M+1)},$$

for some $\xi \in (-x, x)$, for all $x \in [-1, 1]$, where $\psi^{(M+1)}$ is the $(M+1)^{\text{th}}$ order derivative of $\psi$. It follows that, for all $(\mathbf{u}, \mathbf{v}) \in S^n \times S^n$,

$$|k(\mathbf{u}, \mathbf{v}) - k_M(\mathbf{u}, \mathbf{v})| \leq \max_{x \in [-1,1]} |\psi(x) - \psi_M(x)|$$

$$\leq \frac{1}{(M+1)!} \max_{x \in (-1,1)} |\psi^{(M+1)}(x)|.$$

$\square$

The above theorem states that the absolute error made by cutting off the terms beyond order $M$ is less than or equal to the maximum absolute value of the $(M+1)^{\text{th}}$ order derivative of $k$, attenuated by a factor of $1/(M+1)!$. To put this into context, since we use $M = 10$, the attenuation factor is around $2.5 \times 10^{-8}$. Therefore, to make the error significant, the $11^{\text{th}}$ order derivative of the kernel would have to be very high, suggesting a kernel function with abrupt changes looking almost like discontinuities. Since such functions are unlikely to be useful to learn a generalizable model, we believe that the error caused by this approximation is indeed negligible. Note also that, we set $k_M$ to

be the Maclaurin polynomial in the proof of Theorem E.1 to make the derivations easier. However, since we have the freedom to learn the coefficients of the summation, it is theoretically possible to approximate $k$ even better. In particular, even when $k$ has abrupt changes causing a non-negligible error in the Maclaurin approximation, it will be possible to capture some of the residuals using the kernels $k_{\text{odd}}$ and $k_{\text{even}}$ in $k_M$'s expansion. This is because the higher order kernels reach one of these kernels in the limit as discussed in the proof of Theorem 4.3.

# F    COMPARISON TO NON-PARAMETRIC KERNEL METHODS

We use a parametric model with kernels in our method. This is in contrast to the more popular usage of kernels with non-parametric models, such as support vector machines and Gaussian processes. Non-parametric models are usually more flexible. They are also more interpretable than deep network-based parametric models. However, unfortunately, non-parametric models scale poorly with the train set size. For example, the number of support vectors grow linearly with the train set size (Steinwart, 2003), and Gaussian processes methods scale with the cube of the train set size, or linearly after some optimizations (Wilson & Nickisch, 2015).

In contrast, parametric models, such as the one proposed in this work, scale well with the training set size since they use a constant number of parameters regardless of the number of training examples. In fact, this is one of the main reasons why deep learning methods have become extremely popular in the recent years. Non-parametric models also allow faster inference, making them suitable for developing models for compute-limited scenarios, which is the primary focus of this work.

