# OpenReview forum: "Model-Efficient Deep Learning with Kernelized Classification"
_ICLR.cc/2022/Conference — ICLR 2022 Submitted_

### Official Review · Reviewer_PtK6 · 2021-10-27

**Correctness:** 4
**Technical Novelty And Significance:** 2
**Empirical Novelty And Significance:** 2
**Recommendation:** 6
**Confidence:** 3

**Main Review:**

Pros:

The paper is overall well-presented with clear description of the technical methods. The authors conducted extensive experiments with different tasks, the proposed method is effective by outperforming the compared baselines and other methods. Moreover, several ablation study setting are explored and thus strengthen the convincingness.

Cons:
1. The authors conducted the experiment on various ResNet backbone. Instead, it could be more convincing if the author conduct experiment on more types of backbones as in Wang et al. 2019. Besides, the authors have emphasized that comparing with MKL, the proposed method scales better in large dataset setting. I suggest the author conduct experiment on ImageNet like Wang et al. 2019 to show empirical novelty from this perspective.
2. The idea of this work is not quite original. Clearly, this work is inspired by the MKL work and Kervolution neural network. Though the authors have included these work in experiment and show the empirical superiority, the originality and novelty of the idea itself is not very strong.

**Summary Of The Paper:**

The paper presents a kernelized classification layer as an alternative to the linear classification layer that is widely used in deep networks. The kernel classification layer maps learnt feature vector and weight vector to higher dimensional space to conduct scorer function. The key idea is to represent a kernel into a linear combination of predefined kernels and the combination coefficients are jointly learned with the backbone network.The author showed experiment results on multiple tasks to demonstrated the superiority of the proposed layer.

**Summary Of The Review:**

Overall, this paper is well written and the idea is clear. Great amount of experiments are conducted and empirical novelty is shown. Though the originality of the idea is not very strong, I would conclude my current review for this work as: merits over flaws, and marginally above the acceptance threshold.

---

> ### Author Response · Authors · 2021-11-23
> **Author Response to Reviewer PtK6**
>
> We thank the reviewer for the encouraging comments and for recognizing our work as "merits over flaws". We address the reviewer's key concerns below:
>
> > The authors conducted the experiment on various ResNet backbone. Instead, it could be more convincing if the author conduct experiment on more types of backbones as in Wang et al. 2019. Besides, the authors have emphasized that comparing with MKL, the proposed method scales better in large dataset setting. I suggest the author conduct experiment on ImageNet like Wang et al. 2019 to show empirical novelty from this perspective.
>
> We would like to highlight that we conduct experiments on natural language understanding tasks as well. Unlike Wang et al. 2019b, our method is not specific to convolutional networks. We therefore believe that experiments on Transformer-based models in Section 6.3 are also a key component of our experimental evaluation. We would also like to highlight that the focus of our work is to propose a method for boosting the accuracy of deep networks with weak feature extractors. We did not intend to claim state-of-the-art accuracy on image classification benchmarks.
>
> > The idea of this work is not quite original. Clearly, this work is inspired by the MKL work and Kervolution neural network. Though the authors have included these work in experiment and show the empirical superiority, the originality and novelty of the idea itself is not very strong.
>
> A key difference between traditional MKL, kervolutional networks, and our method is that ours is backed by Theorem 4.3, which shows that we optimize over all possible radial positive definite kernels to automatically find the best kernel. Traditional MKL or kervolutional networks, on the other hand, provides no such guarantee. In Section 7.1, we have empirically shown the benefits of Theorem 4.3, where we compare our method to learning the bandwidth parameter of the Gaussian RBF kernels and to learning MKL-like coefficients for a fixed set of kernels. We have also added a new theorem in Appendix E to further justify our usage of Theorem 4.3.

---

### Official Review · Reviewer_ZEPc · 2021-10-30

**Correctness:** 3
**Technical Novelty And Significance:** 2
**Empirical Novelty And Significance:** 2
**Recommendation:** 3
**Confidence:** 4

**Main Review:**

In this paper the authors propose a method that uses nonlinear classification stage at the end of the deep neural network classifiers. To this end, they use kernel trick and radial kernel functions. The strengths and the weaknesses of the proposed method can be summarized as follows:
Strengths:
i) A new method is proposed using radial kernel functions.
ii) The proposed method outperforms the deep neural network methods using softmax loss function and some other methods using kernelized classifiers.
Weaknesses:
i) The main weakness of the paper is that the proposed method is not compared to the related methods well. The authors briefly mention some of the deep neural network classifiers using kernelized classifiers yet there is not any satisfactory argument why the proposed method must be an alternative to these methods. The authors state that the main difference is the use of positive definite kernels (more precisely, they argue that the related methods do not use positive definite kernels whereas they use them). This is unacceptable, in fact majority of the related methods also use positive definite kernels and there should be basic arguments clearly describing the differences and advantages of their proposed method over these ones.
ii) Experimental study is also very weak. The authors only compare their method to Kervo on asingle dataset. But, comparisons to the other deep neural networks using  kernelized classifiers (such as KerNet [R1],  Convolutional Kernel Networks of Marial et al., Deep Kernel [R2], etc.) are missing.
iii) There is a recent study [R3] demonstrating that every deep neural network classifier using standard gradient descent algorithms is in fact mathematically approximately equivalent to kernel machines, a learning method that simply memorizes the data and uses it directly for prediction via a similarity function (the kernel). This somewhat shows that deep neural networks are in fact kernel machines and there is not a need for using nonlinear classifier stage.
iv) I also strongly believe that the proposed method has major shortcomings compared to the kernelized classifiers implemented by SVM type algorithms. Please note that nonlinear SVMs use support vectors that are selected from training samples that lie in critical zones where the class samples approach to each other. Therefore, when evaluating a decision function one has to use multiple kernel function valuation against the support vectors. However, in the proposed method, there is no concept of support vectors. To assign a test sample, we make evaluation only a single vector, w (All m+2 kernel use the same w). This already restricts to return complex decision boundaries. In fact, the same effect can be obtained by using successive fully connected layers.
v) for synthetic experiment, it would be better if the authors show the decision boundaries obtained on a 2d or 3d embedding space returned by deep neural network classifiers as in Center loss paper of Wen et al.

References
[R1] Lauriola et al. “enhancing deep neural networks via multiple kernel learning” Pattern Recognition, 101, 2020.
[R2] Let et al. “Deep Kernel: Learning kernel function from data using deep neural network,” International conference on big Data Computing, Applications and Technologies, 2016.
[R3] Pedro Domingos, “Every model learned by Gradient Descent Is Approximately
a Kernel Machine,” arXiv:2012.00152, 2020.


**Summary Of The Paper:**

In this paper the authors propose a method that uses nonlinear classification stage at the end of the deep neural network classifiers. The authors argue that deep neural networks learn nonlinear representations by using convolutional layers and activation functions yet a linear classifier is used on the learned embeddings which is suboptimal. To overcome this problem, the authors integrate a kernelized classifier that uses a radial kernel function to the last stage of the deep neural network architecture. Then the proposed method is compared to a baseline deep neural network classifier using softmax loss function. Some improvements are obtained compared to the baseline network.

**Summary Of The Review:**

The advantages of the proposed method over the related methods are not discussed and demonstrated well. There is also limitations of the proposed method as described in may main review in the sense that its capacity for creating highly nonlinear decision boundaries is limited. Therefore, I believe the contributions are not enough for acceptance.

---

> ### Author Response · Authors · 2021-11-23
> **Author Response to Reviewer ZEPc - Part I**
>
> We thank the reviewer for the detailed comments and address the key concerns below.
>
> > The main weakness of the paper is that the proposed method is not compared to the related methods well. The authors briefly mention some of the deep neural network classifiers using kernelized classifiers yet there is not any satisfactory argument why the proposed method must be an alternative to these methods.
>
> One of the main advantages of our method compared to existing methods is that we optimize over all possible radial kernels automatically to find the best kernel for the given problem. We also provide rigorous analysis in Theorem 4.3 and the newly added Theorem E.1, that theoretically backs our claim. We believe that these are interesting scientific contributions in addition to the empirical results we have shown. Section 7.1 provides ablation studies to further verify the practical benefits of Theorem 4.3. We have summarized these distinctions of our work when compared with other works in the last paragraph of Section 1 and also in Section 2.
>
> > The authors state that the main difference is the use of positive definite kernels (more precisely, they argue that the related methods do not use positive definite kernels whereas they use them). This is unacceptable, in fact the majority of the related methods also use positive definite kernels and there should be basic arguments clearly describing the differences and advantages of their proposed method over these ones.
>
> We believe that there is a misunderstanding on the part of the reviewer. Please note that our discussion regarding non-positive definite kernels was specific to Wang et al 2019b. We also provided specific examples to the non-positive definite kernels we were referring to ($L^p$-norm kernels in Section 3.3 of that paper). Another distinction of our work compared to Wang et al. is that their method is specific to CNNs. Our method, on the other hand, is more generic and works with other deep network models such as Transformers as demonstrated in Section 6.2. We would also like to reiterate that we are not aware of any other method that provides a theoretical proof that they optimize over all possible radial positive definite kernels to find the best kernel within a deep network.
>
> > Experimental study is also very weak. The authors only compare their method to Kervo on asingle dataset. But, comparisons to the other deep neural networks using kernelized classifiers (such as KerNet [R1], Convolutional Kernel Networks of Marial et al., Deep Kernel [R2], etc.) are missing.
>
> Thanks for the pointers on related papers, we have added citations to them. However, we disagree with the reviewer’s comment on the experiments. Note that we have presented experimental results with image classification with 6 different networks (Section 6.2), 4 natural language understanding tasks & datasets, across 5 different Transformer networks (Section 6.3), knowledge distillation (Section 6.4), active learning (Section 6.5), and a number of ablation studies with various baselines (Section 7). More experiments are available in Appendix as well. It therefore does not seem fair to call the experimental study “very weak”. Note also that Reviewer WDU4 says _"the empirical studies are thorough and the ablation is insightful"_. Reviewer PtK6 says _"authors conducted extensive experiments with different tasks, the proposed method is effective by outperforming the compared baselines and other methods"_.
>
> KERVO in Wang et al, 2019b is applicable only to CNNs, it is therefore not possible to provide that baseline for natural language processing tasks. Note also that we compare against the SOP method in both Section 6.1 and Section 6.2, which is also a kernel method as discussed in Section 2. Also note that Table 6 compares our method to different MKL settings, which are also kernel methods.
>
> KerNET [R1] uses a kernel for _every_ intermediate neural-network feature map and performs traditional MKL. Due to that method not scaling well with the image size and the number of intermediate representations in the network, the original paper reports results on MNIST images (784 pixels) with a 1 to 6 layer plain, LeNet-style CNN. Since the source code of the method is not publicly available, it is not trivial for us to implement it for a setting which is much more complicated than what the original authors of the paper used (CIFAR images have 3072 pixels and ResNet networks have much more complicated internal representations than LeNet style CNNs KerNET used).
>
> Deep Kernel of [R2], is a binary classification model that takes a pair of data points (images) as inputs and outputs a binary classification. It also uses Deep Belief Networks and Restricted Boltzmann Machines instead of CNNs/Transformers we use. Therefore, it is unclear how the work in [R2] can be adapted to be a meaningful baseline for a fair comparison with our work. Note that the source code for this method is not publicly available either.

---

> > ### Author Response · Authors · 2021-11-23
> > **Author Response to Reviewer ZEPc - Part II**
> >
> > > There is a recent study [R3] demonstrating that every deep neural network classifier using standard gradient descent algorithms is in fact mathematically approximately equivalent to kernel machines, a learning method that simply memorizes the data and uses it directly for prediction via a similarity function (the kernel). This somewhat shows that deep neural networks are in fact kernel machines and there is not a need for using nonlinear classifier stage.
> >
> > While we appreciate this argument and reference [R3] on seeing deep neural network classifiers as kernel machines, this does not diminish our contribution in any way. We provide an example to prove our point: There are theoretical results showing that MLP networks are universal approximators. In light of these results, one could question the usefulness of having specialized architectures like CNNs and Transformers over using plain MLPs. However, inductive-bias based models such as CNNs and Transformers have been immensely successful in solving problems better than MLPs do. Kernelized classification can also be viewed as a way of presenting an inductive bias motivated by the classic kernel method theory. We have discussed this in Section 7.4 of the paper.
> >
> > > I also strongly believe that the proposed method has major shortcomings compared to the kernelized classifiers implemented by SVM type algorithms. .… In fact, the same effect can be obtained by using successive fully connected layers.
> >
> > We agree with the reviewer that non-parametric models, such as SVMs, are more flexible than parametric models. We have added a new section, Appendix F, to discuss the advantages and disadvantages of parametric vs non-parametric models. The primary advantage of using a parametric model like we do is that it scales well with the training set size. In SVM, the number of the support vectors, and therefore, the storage requirement for the model, and the inference time grow linearly with the training dataset size [Steinwart, 2003]. Our model’s size and the inference time remain constant with the training dataset size, making it suitable for compute/memory-limited applications, which is the primary focus of our work.
> >
> > Regarding the number adding more fully connected layers to obtain complex decision boundaries, we did provide a comparison with this baseline in Section 7.4. As discussed in that section, although MLPs can theoretically approximate any function, learning one from data is difficult. This has motivated inductive-bias based models such as CNNs and Transformers. Kernelized classification is also a way of providing an inductive bias motivated by theory discussed in the paper. Note also that, unlike the kernelized classification layer, added MLP layers come with a significant increase in the model’s computational and memory complexity.

---

> > ### Comment · Reviewer_ZEPc · 2021-11-23
> > **clarification**
> >
> > Regarding the weakness of the experimental study, I had better emphasized that it is weak in the sense of comparison to other related methods as described in my review. In terms of the tested datasets, I agree with you, it is pretty rich in this sense.

---

> > > ### Author Response · Authors · 2021-11-30
> > > **Response to Reviewer ZEPc**
> > >
> > > Thanks for the clarification. We would like to highlight that we compare our method to kervolution, second-order pooling, linear kernel, automatic bandwidth parameter tuning of the Gaussian RBF kernel, and a traditional-MKL-like setting with multiple pre-defined kernels, all of which are kernel-based baselines useful to critically evaluate the strengths/weaknesses of the proposed method.
> > >
> > > We appreciate the references [R1] and [R2] and have included them in the updated version of the paper. To summarize why we are unable to provide an empirical comparison to these methods:
> > >
> > > [R1] - As described in Section 4.2 of [R1], they use MKL with a kernel for every intermediate feature map, and perform a grid search with the number of CNN layers. The authors only consider a straightforward LeNet-style CNN with 1 to 6 layers. Implementing the same method in our setting with ResNet networks consisting up to 56 CNN "blocks" is a much more complicated task (note that the actual number of intermediate layers in a ResNet network is much higher than the number of "blocks"). It is a much harder task than what the original authors of [R1] undertook, both in terms of the effort required to come up with an efficient enough implementation to handle a large number of kernels and the resources required for experimentation.
> > >
> > > [R2] - This model takes a pair of images as inputs and outputs a binary classification score. It uses Deep Belief Networks and Restricted Boltzmann Machines instead of CNNs/Transformers that we use. Therefore, it is unclear how the work in [R2] can be adapted to the multi-class supervised classification setting to be a fair baseline for our method.

---

### Official Review · Reviewer_WDU4 · 2021-10-31

**Correctness:** 4
**Technical Novelty And Significance:** 3
**Empirical Novelty And Significance:** 3
**Recommendation:** 6
**Confidence:** 4

**Main Review:**

# Strengths
- The paper is well-written and well-motivated.
- The problem the paper is concerned with is interesting. Instead of widening or deepening the model, the authors determine to use a more powerful classification layer for parameter efficiency.
- The empirical studies are thorough and the ablation is insightful.


# Weaknesses & concerns
- Your assumption "this is suboptimal since better nonlinear classifiers could exist in the same embedding vector space" perhaps should only apply to the backbones with limited expressiveness, e.g., the used ResNet-CIFAR architectures. As you have not tried more powerful architectures like the wide-ResNet family or the ResNet-ImageNet family, I hold concerns about the correctness of this assumption.

- You claim "we theoretically show that our classification layer optimizes over all possible kernel functions on the space of embeddings to learn an optimal nonlinear classifier". But, as detailed in Sec 4, you confine the kernel family as the radial kernels. You can clarify this in the original statement to avoid over-claim.

- When first hearing the idea of combining DNNs and kernelized classifiers, I thought that the non-parametric kernelized classifiers are leveraged to make predictions. But, in fact, you insist on the parameterized softmax classifier (Eq 3). Can you provide a discussion on the pros and cons of the two kinds of model choices? I realize the primary con of non-parametric kernelized classifiers is that they are not so scalable in the test phase, although they enable analytical inference.

- If you write the feature extractor as a mapping f and combine f with the radial kernel layers, you are basically defining a deep kernel. The main difference between this work and Wilson et al., 2016 is that you normalize the output of f and learn with parameterized modeling as well as SGD?

**Summary Of The Paper:**

The paper considers replacing the linear softmax layer at the end of a classifier with a kernelized softmax layer to boost the expressiveness of the whole model, especially when using a lightweight feature extractor.
The authors have provided theoretical justifications for their layer design, and claim that the designed kernelized softmax layer optimizes over all possible kernel functions on the space of embeddings.
The authors evidence the usefulness of this layer in learning more model-efficient classifiers in a number of computer vision and natural language processing tasks.

**Summary Of The Review:**

Given the identified weaknesses, I recommend a weak acceptance for this submission. I am glad to increase my score if the authors are able to resolve my concerns.

---

> ### Author Response · Authors · 2021-11-23
> **Author Response to Reviewer WDU4**
>
> Thanks for the detailed comments. We are glad that the reviewer receives this work positively. We address the reviewer's concerns below:
>
> > Your assumption "this is suboptimal since better nonlinear classifiers could exist in the same embedding vector space" perhaps should only apply to the backbones with limited expressiveness, e.g., the used ResNet-CIFAR architectures. As you have not tried more powerful architectures like the wide-ResNet family or the ResNet-ImageNet family, I hold concerns about the correctness of this assumption.
>
> We agree with the reviewer. This work certainly focuses on backbones with limited expressiveness, which have their own applications in memory/compute-limited settings as discussed in Section 1. We have revised the above sentence as: “This could be suboptimal for a network with a limited-capacity backbone since better nonlinear classifiers could exist in the same embedding vector space”. We also start the abstract by saying “We investigate the possibility of using the embeddings produced by a lightweight
> network more effectively with a nonlinear classification layer.”
>
> > You claim "we theoretically show that our classification layer optimizes over all possible kernel functions on the space of embeddings to learn an optimal nonlinear classifier". But, as detailed in Sec 4, you confine the kernel family as the radial kernels. You can clarify this in the original statement to avoid over-claim.
>
> We have revised this claim as suggested by the reviewer to “We theoretically show that our classification layer optimizes over all possible radial kernel functions on the space of embeddings to learn an optimal nonlinear classifier”.
>
> We would also like to mention that all the positive definite kernels that have been used successfully on the sphere, including, polynomial kernels, squared-exponential (Gaussian) kernels, Laplacian kernels, are radial kernels. In fact, we are not aware of any non-radial kernel on the sphere that has been used before in the mainstream machine learning literature.
>
> > When first hearing the idea of combining DNNs and kernelized classifiers, I thought that the non-parametric kernelized classifiers are leveraged to make predictions. But, in fact, you insist on the parameterized softmax classifier (Eq 3). Can you provide a discussion on the pros and cons of the two kinds of model choices? I realize the primary con of non-parametric kernelized classifiers is that they are not so scalable in the test phase, although they enable analytical inference.
>
> Following the reviewer’s suggestion, we have added a new section (Appendix F) with pros and cons of the two model choices. In summary, non-parametric models such as SVMs and GPs are more flexible and more interpretable. However, unfortunately, they scale poorly with the training set size (e.g. the number of support vectors grows linearly with the number of training examples [Steinwart, 2003]). Since our work focuses on improving the accuracy of light-weight models used in compute/memory-limited settings, parametric-models are more appropriate for our case because the model size and inference time both have O(1) complexity.
>
> (Steinwart, 2003) Ingo Steinwart. Sparseness of Support Vector Machines. JMLR, 2003.
>
> > If you write the feature extractor as a mapping f and combine f with the radial kernel layers, you are basically defining a deep kernel. The main difference between this work and Wilson et al., 2016 is that you normalize the output of f and learn with parameterized modeling as well as SGD?
>
> We agree with the differences highlighted by the reviewer. Some other differences between our work and Wilson et al. 2016 are:
>
>
> * Our work focuses on the classification setting using the categorical cross entropy loss, whereas Wilson et al. focuses on the regression setting with a Gaussian Processes formulation. We plan to extend our work to the regression setting in the future.
>
> * Wilson et al. use Gaussian RBF and spectral mixture kernels. Our method has the capability to automatically learn _any_ positive definite radial kernel. Note that Gaussian RBF and spectral kernels are all radial kernels.
>
> * Wilson et al. is a non-parametric model, which scales linearly with the number of training examples. In contrast, our model, being a parametric model, scales O(1) with the number of training examples. These two different kinds of model have pros and cons discussed in Appendix F.
>
> * Wilson et al. uses a two-stage training procedure, first training the neural network part with the MSE loss and then further tuning all the parameters with a KISS-GP model. Over method, however, is trained in one go with the standard deep network training tools.

---

> > ### Comment · Reviewer_WDU4 · 2021-11-29
> > **Reply to the rebuttal**
> >
> > I thank the authors for the clarification. I am satisfied with the provided explanations. Nevertheless, is it better to emphasize in the next version that your main technical contribution is using deep radial kernels for parametric classification?

---

> > > ### Author Response · Authors · 2021-11-30
> > > **Response to Reviewer WDU4**
> > >
> > > Thanks for the suggestion. We will emphasize this in the next version.

---

### Official Review · Reviewer_pWF3 · 2021-11-05

**Correctness:** 2
**Technical Novelty And Significance:** 2
**Empirical Novelty And Significance:** 2
**Recommendation:** 3
**Confidence:** 4

**Main Review:**

The paper is generally well written with the ideas explained
clearly. However, the main problem I have is the generality of this
approximation used to define a functional for evaluating the
kernel. The paper truncates higher order terms in the kernel expansion
and fixes the number of terms (M) to 10 in all the experiments. The
math however shows that the kernel is indeed sensitive to M, which
isn't surprising. The experiment in Figure 2 doesn't establish
anything significant in this regard. The paper will benefit immensely
from a more rigorous treatment of the kernel approximation.


**Summary Of The Paper:**

This work considers the problem of using a non-linear classification
function as a final layer of a neural network instead of the
conventional softmax function. To this end, the authors proposed a new
kernelized classification layer which is approximately learned using a
modified version of the standard softmax classification
loss. Experimental evidence shows that the proposed kernelized
classification layer is able to outperform baselines, especially
standard softmax on synthetic and real world datasets.


**Summary Of The Review:**

The proposed kernel classification layer is a promising idea, however
I believe it needs more theoretical and empirical analysis to make it
generally applicable.

---

> ### Author Response · Authors · 2021-11-23
> **Author Response to Reviewer pWF3**
>
> We thank the reviewer for this valuable suggestion and completely agree that such a treatment is beneficial. We have now added a rigorous analysis of the kernel approximation error to Appendix E of the updated paper, where we introduce and prove the following theorem.
>
> Let $k : S^n \times S^n \to \mathbb{R}$ be any positive definite radial kernel on $S^n$ with the series expansion $k(\mathbf{u}, \mathbf{v}) = \alpha_{-2}k_{\rm even}(\mathbf{u}, \mathbf{v}) +  \alpha_{-1}k_{\rm odd}(\mathbf{u}, \mathbf{v}) + \sum_{m=0}^\infty \alpha_m \langle \mathbf{u}, \mathbf{v}\rangle^m$, and  $k_M : S^n \times S^n \to \mathbb{R}$ be its $M^{\text{th}}$ partial sum.
> Define $\psi: [-1, 1] \to \mathbb{R}$ as $\psi(x) := \sum_{m=0}^\infty \alpha_m x^m$. Then the approximation error bound for the partial sum of the kernel is given by
> $$
>     |k(\mathbf{u}, \mathbf{v}) - k_M(\mathbf{u}, \mathbf{v})| \le \frac{1}{(M+1)!}\max_{x \in (-1, 1)} |\psi^{(M+1)}(x)|,
> $$
> for all $(\mathbf{u}, \mathbf{v}) \in S^n \times S^n$.
>
>
> Our proof is based on the Lagrange remainder of the Maclaurin expansion approximation for an analytic function. The result shows that the approximation error is bounded from above by the absolute value of the 11-th order derivative of the continuous component of $k(\langle x, y \rangle)$ in (-1, 1), _attenuated by a factor of 2.5 x 10^(-8)_. Therefore, the approximation error would be negligible in most cases (specially when working on a computer with limited precision). To make the error significant, the 11-th order derivative of the kernel would have to have an _extremely high_ value in the interval (-1, 1), suggesting a function with abrupt changes looking almost like discontinuities. Such kernels are unlikely to be useful in learning generalizable models.
>
> Note also that, we set $k_M$ to be the Maclaurin polynomial of $k$ in the proof of Theorem E.1 to make the derivations simpler. However, since we have the freedom to learn the coefficients of the summation, it is theoretically possible to approximate $k$ even better. In particular, even when $k$ has abrupt changes causing a non-negligible error in the Maclaurin approximation, it will be possible to capture some of the residuals using the kernels $k_\text{odd}$ and $k_\text{even}$ in the finite summation. This is because the higher order component kernels reach one of these kernels in the limit as discussed in the proof of Theorem 4.3.
>
> We believe that this new theoretical analysis, coupled with the empirical results in Figure 2, provides strong evidence that the kernel approximation is reasonable.

---

### Decision · Program_Chairs · 2022-01-20

**Decision:**

Reject

**Comment:**

The paper proposes the replacement of the softmax layer in a neural network with one parametrized by a kernel. The kernel itself is learned during training from the space of radial basis kernels. The resulting models are compared against identical networks with softmax, linear kernels, second order pooling and kervolutions on several datasets, encompassing vision and NLP tasks.

First, the reviewers raised questions about the novelty of the work. Theorem 4.3, based on which the method is derived, has existed in the literature and seems to be related to the uniqueness of the power series expansions for kernels. There is novelty in using this theoretical result to write an approximation of a positive definite kernel in a way which can be learned. Specifically, it is written as a finite weighted sum of existing kernels, where the coefficients are learned. Reviewer pWF3 posed a valid question about the quality of the approximation, to which the authors responded with an equally valid, and comprehensive, appendix on the error bounds of the approximation. Still, it is worth tempering the statement that the search is 'exhaustive' over the space of radial kernels or that the kernel is optimal (instead, the search appears over a large class of radial kernels, and the kernel is approximately optimal with an extremely low distance from the actual optimum).
Along the same lines of rephrasing claims, reviewer WDU4 also pointed out several statements and claims which were not entirely accurate, which the authors then proceeded to resolve, resulting in notable changes from the initial version of the paper. Specifically, there was mention of a "non-parametric kernelized classifier". This has been fixed, but it did seem to have initially confused other reviewers, who suggested related work that, it turns out, are not necessarily suitable contenders. The changes made definitely improved the paper, and resolved most of the reviewer's concerns.
Nevertheless, the appendix added comparing the method to non-parametric models could be improved. For instance the authors stated "Wilson et al. use Gaussian RBF and spectral mixture kernels. Our method has the capability to automatically learn any positive definite radial kernel. Note that Gaussian RBF and spectral kernels are all radial kernels." - is there any intuition, or proof, of a case when the method introduced here learns a network + classifier that the method by Wilson et al. cannot learn? Or for which deep kernel learning requires considerably more resources? (DKL has been optimized and made considerably faster since the initial paper in 2016).  https://proceedings.neurips.cc/paper/2016/hash/bcc0d400288793e8bdcd7c19a8ac0c2b-Abstract.html
Also, while the present work is backed by 4.3, DKL also has a theoretical grounding.
https://www.jmlr.org/papers/volume20/17-621/17-621.pdf

There was some discussion on the exhaustiveness of the experiments, and it was concluded that the datasets are sufficient, while the reviewers were not in agreement as to whether the authors considered sufficient contenders. A comparison against DKL, at least, appears to be warranted.

Overall, the paper brings a contribution in terms of improving the performance of backbones with limited expressiveness through the use of a kernel-parametrized classifier, learned by optimizing an approximation of a formulation that spans the entire space of radial basis kernels. The paper was updated considerably during the reviewer process, to its betterment, however, an experimental comparison against deep learning with non-parametric kernelized classifiers is still missing.